# The Role of Gut Microbiome on Glioblastoma Oncogenesis and Malignant Evolution

**DOI:** 10.3390/ijms26072935

**Published:** 2025-03-24

**Authors:** Zaynab Sidi Mohamed, Qiong Wu, Maria A. Jacome, Jianan Chen, Arnold B. Etame

**Affiliations:** 1School of Medicine, Tulane University, New Orleans, LA 70112, USA; zaynab@tulane.edu; 2Department of Neuro-Oncology, H. Lee Moffitt Cancer Center & Research Institute, Tampa, FL 33612, USA; qiong.wu@moffitt.org (Q.W.); jianan.chen@moffitt.org (J.C.); 3Department of Immunology, H. Lee Moffitt Cancer Center & Research Institute, Tampa, FL 33612, USA; maria.jacomellovera@moffitt.org

**Keywords:** glioblastoma, gut microbiome, gut–brain axis, therapeutic resistance

## Abstract

Glioblastoma (GBM) remains the most aggressive primary brain tumor, with poor survival outcomes and treatment limited to maximal safe surgical resection, chemotherapy with temozolomide, and radiotherapy. While immunotherapy and targeted treatments show promise, therapeutic resistance and disease progression remain major challenges. This is partly due to GBM’s classification as a “cold tumor” with low mutational burden and a lack of distinct molecular targets for drug delivery that selectively spare healthy tissue. Emerging evidence highlights the gut microbiota as a key player in cancer biology, influencing both glioma development and treatment response. This review explores the intersectionality between the gut microbiome and GBM, beginning with an overview of microbiota composition and its broader implications in cancer pathophysiology. We then examine how specific microbial populations contribute to glioma oncogenesis, modulating immune responses, inflammation, and metabolic pathways that drive tumor initiation and progression. Additionally, we discuss the gut microbiome’s role in glioma therapeutic resistance, including its impact on chemotherapy, radiotherapy, and immunotherapy efficacy. Given its influence on treatment outcomes, we evaluate emerging strategies to modulate gut flora, such as probiotics, dietary interventions, and microbiota-based therapeutics, to enhance therapy response in GBM patients. Finally, we address key challenges and future directions, emphasizing the need for standardized methodologies, mechanistic studies, and clinical trials to validate microbiota-targeted interventions in neuro-oncology. By integrating gut microbiome research into GBM treatment paradigms, we may unlock novel therapeutic avenues to improve patient survival and outcomes.

## 1. Introduction

Glioblastoma (GBM) is the most lethal form of brain cancer. An aggressive grade 4 brain tumor of glial origin, GBM represents the deadliest form among the six glioma families and is characterized by its poor prognosis [1]. Patients typically succumb to the disease within a few years of diagnosis [2]. GBM is characterized by aggressive invasion, cellular heterogeneity, and resistance to conventional therapies [1]. Its median survival time is around 12–16 months, with a 5-year survival rate of only about 5–7% [3,4]. The blood–brain barrier (BBB) significantly limits drug delivery, allowing only about 20% of systemic drug concentrations to reach the brain, thus complicating treatment [3]. Currently, the standard treatments—surgical resection, radiotherapy, and chemotherapy with temozolomide (TMZ)—are often insufficient due to the BBB that restricts effective drug delivery [5]. Additionally, GBM’s capacity to evade immune responses contributes to its resistance to therapies, making new approaches vital [6,7].

The gut microbiome has recently been closely connected to human health and disease including neurophysiology [8]. Recent neurophysiology studies suggest that gut dysbiosis may influence GBM development through immune modulation and metabolic changes. For instance, specific gut microbiota have been associated with GBM risk, possibly affecting immune responses critical to tumor growth [9]. Additionally, research highlights how gut microbiota alterations can impact GBM’s immune environment, potentially affecting tumor progression [10,11]. The gut microbiome might play a critical role in GBM pathogenesis, therapeutic resistance, and the potential to modulate treatment responses, underscoring the urgency of understanding its interaction with the gut microbiome [9,12].

In this review, we first provide an overview of the gut microbiome and its role in cancer pathophysiology. We then examine its specific contributions to glioma oncogenesis, including its effects on T cell dysfunction, tumor-associated macrophages (TAMs) and microglia, and immune modulation in GBM progression. We further explore how gut microbiota influence GBM therapeutic resistance and discuss their potential as therapeutic targets. Finally, we address the challenges and future directions in leveraging gut microbiota for GBM treatment.

## 2. Overview of Gut Microbiome

The gut microbiome has been increasingly associated with various aspects of human health and disease, including immune regulation and metabolic functions [13]. According to the hygiene hypothesis, gut bacteria may influence the development of multiple diseases, such as allergies and autoimmune disorders. Studies have revealed that certain bacterial species are protective against disease onset, while others are linked to accelerating disease progression [14]. The gut is a complex ecosystem consisting of six most dominant bacterial phyla: Firmicutes, Bacteroidetes, Actinobacteria, Proteobacteria, Fusobacteria, and Verrucomicrobia in which Firmicutes and Bacteroidetes are the most prevalent [15]. Genetic tools such as 16S rRNA gene sequencing have facilitated the study of human feces and its genetic composition, establishing a causal link between microbiome health and disease. Other technologies like metagenomic sequencing and metabolomics have expanded beyond 16S rRNA, enabling more precise analysis of gut microbiota functions and their contributions to health and disease [3]. Metagenomics allows researchers to identify microbial genes and their potential roles in metabolic pathways, which are crucial for understanding complex diseases like cancer.

Environmental factors such as diet, antibiotic use, and delivery method at birth significantly shape microbiome composition. In addition to dietary habits, the mode of delivery (vaginal birth vs. C-section) has a long-term impact on an individual’s microbiota [16]. Studies have shown that early exposure to specific microbes can shape immune system responses, possibly influencing the risk of developing autoimmune conditions or certain types of cancer [3]. For example, studies comparing germ-free versus specific pathogen-free mice showcase differences in disease susceptibility, highlighting the role of gut microbiota in immune development [17]. A healthy microbiome, associated with Firmicutes and Bacteroidetes, supports nutrient metabolism, producing short-chain fatty acids (SCFAs) like butyrate, propionate, and acetate, which have anti-inflammatory properties [18]. Moreover, the gut flora protects against pathogenic microbes by maintaining a two-tiered mucus barrier and working symbiotically with intestinal dendritic cells to produce secretory IgA, preventing the translocation of bacteria into the bloodstream.

Importantly, gut bacteria play a critical role in modulating innate and adaptive immune responses, influencing the development of immune cells like Tregs, IgA-producing B cells, and innate lymphoid cells [19]. Bacterial signals, including SCFAs and TLR-MyD88 signaling, help maintain immune balance and fight pathogens [17]. Given the vital role of gut bacteria in maintaining health, disruptions in this balanced relationship can lead to various disorders, including inflammatory bowel diseases (IBDs), cardiovascular issues, neurological disorders, cancers, and disturbances in the gut–brain axis.

The gut–brain axis represents a complex, bidirectional communication system influenced by various factors such as diet, exercise, medications, stress, and overall well-being [20]. Research in mouse models demonstrates that gut microbiota can significantly affect neurophysiology. This bidirectional communication network is mediated by multiple pathways involving the autonomic nervous system (ANS), enteric nervous system (ENS), central nervous system (CNS), immune system, and endocrine system. It regulates neuroinflammation, neurotransmission, and even neurogenesis. The ANS plays a crucial role by mediating gut responses and facilitating interactions between the gut microbiota and ENS. Furthermore, gut microbiota can influence the CNS through metabolites like serotonin, GABA, and indole, as well as interact with the hypothalamic–pituitary–adrenal (HPA) axis to modulate stress responses [20].

For example, studies have shown that feeding mice *Lactobacillus rhamnosus* upregulated GABA gene expression in brain regions such as the amygdala, hippocampus, and locus coeruleus, indicating the microbiota’s direct impact on brain function [21]. Additionally, another study revealed that transferring gut microbiota from Parkinson’s disease patients into mice induced motor deficits, microglial activation, and alpha-synuclein pathology in the mice, suggesting a direct link between gut health and neurodegenerative conditions [22]. These findings suggest that the gut microbiome can influence neurological conditions, potentially extending to neuro-oncological diseases like glioma. The bidirectional interaction between the gut–brain axis has attracted considerable interest recently, yet the connection between the gut microbiome and neuro-oncological conditions, particularly gliomas, remains underexplored [23].

## 3. Gut Microbiome in Cancers

The link between the gut microbiome and various cancers is emerging as a significant area of study. Specific bacteria like *Helicobacter pylori* have well-documented roles in gastric cancer, while *Bacteroides fragilis* is linked to colorectal cancer progression [24]. Emerging evidence suggests that gut microbiota may modulate immune responses, potentially affecting lung cancer development and response to therapies [25]. Gut microbiota can also impact estrogen metabolism, thereby influencing breast cancer risk and progression [26]. Understanding these connections has paved the way for investigating similar roles in GBM [3]. Bacterial metabolites significantly influence immune checkpoint pathways, impacting cancer immunotherapy outcomes. SCFAs like butyrate, produced by gut bacteria such as *Bacteroides fragilis*, can influence immune checkpoint molecules by modulating regulatory T cells (Tregs). Butyrate, in particular, enhances the expression of CTLA-4 (cytotoxic T-lymphocyte-associated protein 4) on Tregs, which suppresses immune responses and promotes an immunosuppressive environment that may affect tumor growth and immune tolerance [27]. Additionally, certain gut bacteria convert primary bile acids into secondary bile acids, such as deoxycholic acid, which interact with the farnesoid X receptor (FXR) and G-protein-coupled bile acid receptor 1 (TGR5) on immune cells. This interaction can aid immune escape in tumor microenvironments [28]. Certain bacterial metabolites can influence immune checkpoint pathways like PD-L1, potentially affecting the immune environment within tumors, including brain cancers [4]. These findings are relevant to the hypothesis that modulating the gut microbiome could alter the immune response in GBM.

Studies using mouse models have shown significant differences in the gut microbiota composition between glioblastoma-bearing mice and healthy controls, suggesting a potential role of the microbiome in tumor progression [29]. The gut microbiome may play a role in the development of glioma by modulating immune responses and the microenvironment of the CNS [29]. A study analyzing GBM tissue composition identified the presence of 22 distinct bacterial taxa components mainly inside tumor cells within the tumor microenvironment, further supporting the notion that bacteria could play a role in glioma biology [30]. Furthermore, preliminary research in GBM patients reveals increased microbial diversity and shifts in specific bacterial species, suggesting a state of gut dysbiosis. Specifically, increased levels of Proteobacteria and decreased Firmicutes are observed in GBM patients compared to healthy controls at the phylum level. At the family level, Enterobacteriaceae, Bacteroidaceae, and Lachnospiraceae are elevated, while Ruminococcus, Faecalibacterium, and Prevotella are reduced. Finally, at the species level, *Bacteroides vulgatus* and *Escherichia coli* are notably elevated in the GBM group [23].

In another study, qualitative and quantitative analyses have shown significant differences in gut microbiota between GBM patients and healthy controls [31]. For instance, Enterobacteriaceae has been identified as one of the most prevalent bacterial families across various cancer types, highlighting potential bacteria–tumor associations [31]. Additionally, Lactobacillus genera are significantly decreased in GBM patients. This reduction is notable because Lactobacillus is involved in maintaining selenium levels, which are often deficient in patients with brain cancers. This finding further supports the role of gut bacteria in influencing glioblastoma metabolism and progression.

## 4. The Role of Gut Microbiome in Glioma Development (Oncogenesis)

GBM is characterized by microvascular proliferation, cellular heterogeneity, bilateral invasion, and pseudopalisading necrosis, all contributing to its complexity and resistance to treatment [1]. Glioma progression is strongly influenced by its ability to create a complex immunosuppressive microenvironment that supports tumor survival [6]. This environment promotes the growth of host immune cells like microglia and monocytes, which constitute about 30% of the tumor mass, ultimately decreasing patient survival chances [32,33,34]. The immune landscape of GBM is further shaped by tumor-associated macrophages (TAMs), which adopt an immunosuppressive M2-like phenotype under the influence of gut microbiota metabolites, aiding tumor growth [35]. In the following sections, we first review the various immune mechanisms that promote GBM progression, and we finally explore the contribution of the gut microbiome in tumorigenesis.

### 4.1. GBM’s Effects on T Cells: Inducing Immune Dysfunction

A functioning, adaptive immune system, particularly an effective T cell response, is crucial for mounting an anti-tumorigenic defense. Regulatory T cell expansion, T cell senescence, and T cell apoptosis are different targets of T cell dysfunction. Fornatly et al. demonstrated that CD4+ T cells in GBM patients exhibit immunosenescence, marked by a significant increase in CD4+CD28-CD57+ T cells, a phenotype associated with T cell replicative senescence [36]. CD57 is recognized as a marker of terminal T cell differentiation, while CD28 is a co-stimulatory marker necessary for T cell activation [37]. GBM tumor cells exploit immune tolerance mechanisms, which are typically in place to prevent autoimmunity or responses to innocuous antigens such as food-derived peptides. Notably, GBM induces regulatory T cell (Treg) production or triggers peripheral deletion of inflammatory responses to evade T cell-mediated attacks. Peripheral deletion involves increased Fas ligand expression on CD4+ and CD8+ T cells, signaling apoptosis through caspase and endonuclease activation [38].

GBM patients demonstrate elevated levels of CD4+ Tregs both systemically and within the tumor microenvironment, exemplified by CD4+CD25+FOXP3+CD45RO+ T cell phenotypes. Removal of these Tregs in vitro has been shown to restore T cell proliferation and reverse the Th2 cytokine shift characteristic of Treg phenotypes [39]. GBM facilitates this Treg-heavy phenotype by upregulating immunosuppressive molecules such as indoleamine 2,3-dioxygenase (IDO) in dendritic cells and the chemokine CCL2, which attracts Tregs [40]. Additionally, GBM-derived macrophages express T cell immunoglobulin and mucin domain-containing molecule 4 (TIM4), which can phagocytose tumor-specific T cells while increasing the expression of immunosuppressive cytokine TGFβ, further promoting Treg induction [41]. The transcription factor STAT3 is also implicated in promoting GBM tumor survival, proliferation, and invasion [42].

Furthermore, GBM evades T cell infiltration and attack by downregulating major histocompatibility complex class I (MHC I), which is critical for antigen presentation and CD8+ cytotoxic T cell activation [43]. Simultaneously, GBM upregulates co-inhibitory molecules such as PD-L1 and CTLA-4, facilitating Treg recruitment and immune suppression [44,45]. These mechanisms underscore how GBM reprograms the immune microenvironment to promote tumor survival by impairing cytotoxic T cell activation. Furthermore, inflammation within the tumor microenvironment plays a central role in shaping immune cell behavior, influencing the balance between tumor-promoting and tumor-suppressing processes. Understanding this interplay between inflammation, immune evasion, and the role played by gut flora highlights potential therapeutic strategies to restore anti-tumor immunity.

Naghavian et al. demonstrated that bacteria and gut microbiota are present within GBM, with their peptides being presented by HLA molecules on GBM cells and local antigen-presenting cells (APCs) [46]. These microbial peptides, in turn, stimulate tumor-infiltrating lymphocytes (TILs) [46]. This cross-reactivity suggests a potential link between cancer responses, bacterial peptides, and immune checkpoint inhibitors, highlighting therapeutic opportunities that leverage T cell cross-reactivity against bacterial antigens and tumor peptides. The microbiome also affects the balance between pro-inflammatory Th1/Th17 cells and regulatory T cells (Tregs). *Bacteroides fragilis* promote Tregs that contribute to localized immunosuppression, facilitating tumor progression in GBM [47].

### 4.2. Microglia and M2 Macrophage Polarization and Tumor Growth

The microbiome can influence immune cells like microglia, T cell subsets, and dendritic cells, which play key roles in tumor progression and immune modulation [48]. Tumor-associated macrophages (TAMs) and Tregs create a feedback loop, where TAM-derived TGF-β supports Tregs, which in turn further suppress effector T cells. TAMs and microglia represent a significant component of the GBM microenvironment. These myeloid-derived cells, instead of mounting an anti-tumor response, are often polarized into an immunosuppressive M2 phenotype, which aids tumor progression. Studies using the GL261 syngeneic mouse model of glioma have shown that treatment with specific antibiotics alters gut microbiota composition, which in turn affects immune responses, such as reducing cytotoxic NK cell subsets while paradoxically increasing other NK cells and microglial phenotypes [49]. These alterations were associated with the subsequent onset of glioma in the treated mice, suggesting a role for gut microbiota in tumor development [49]. A higher ratio of M2 to M1 microglia is correlated with poorer outcomes, as M2 microglia tend to suppress T cell responses and promote tumor growth. Ruminococcaceae might help shift this balance through the production of the metabolite isoamylamine, which could induce microglial cell death or alter their polarization. This shift might promote M1-type pro-inflammatory activity while reducing the tumor-supportive role of M2 cells, suggesting that targeting Ruminococcaceae in the gut could enhance immune responses against GBM and open new therapeutic pathways [50]. Additionally, Prevotella7 has been identified as a producer of alpha-galactosylceramide, a metabolite that activates invariant natural killer T cells, thus promoting anti-cancer immune activity [51]. This aligns with findings from Zeng et al., suggesting a potential protective role for Prevotella7 in GBM. Prevotella7 has also been shown to confer protection against colorectal cancer, indicating its broader relevance in cancer prevention [9,52].

Glioma progression correlates with shifts from Firmicutes-dominated microbiota in healthy mice to Bacteroides dominance in glioma-bearing mice. Gut-derived metabolites such as tryptophan, arginine, glutamate, glutamine, and SCFAs have significant roles in glioma development [47]. Tryptophan metabolites can activate pathways affecting immune responses and tumor growth, while arginine impacts tumor cell proliferation. Additionally, glutamate and glutamine are critical for energy metabolism, and SCFAs contribute to immune regulation and microglial function, influencing the tumor microenvironment and the BBB [47].

Overall, immune failure in GBM occurs at the level of several key cell players including T cells, microglia, TAMs, and NK cells (Figure 1).

### 4.3. Gut Microbiota’s Role in Modulating Immunity and GBM Progression

Bacterial metabolites like indole derivatives and SCFAs (e.g., butyrate) can influence systemic inflammation and immune responses, potentially playing a role in glioma initiation and progression [53]. SCFAs, known for their influence on neurophysiological processes, can impact brain function, as highlighted in research linking gut microbiota to behavioral changes [53]. Specifically, changes in SCFAs during glioma development have been linked to altered immune regulation in mouse models [4]. Dono et al. demonstrated that glioma affects the levels of SCFas propionate, butyrate, and acetate. These metabolites, critical for maintaining gut–brain homeostasis, are all decreased after glioma development in both murine glioma models and humans [12]. Changes in metabolites, including reduced norepinephrine and serotonin levels, underscore how gut dysbiosis could contribute to glioma development through altered neurotransmission and immune suppression. The neurotransmitter metabolites 5-hydroxyindoleacetic acid (5-HIAA) and norepinephrine showed reductions. Interestingly, long-term use of tricyclic antidepressants has been associated with a lower incidence of glioma, suggesting a potential protective effect [54]. Furthermore, studies have shown that dopamine and serotonin levels correlate positively with tumor burden in GBM patients [55]. Specifically, serotonin levels were found to increase following glioma onset.

Recent studies have demonstrated that temozolomide (TMZ) treatment can modulate the alterations in fecal metabolites induced by glioma. For instance, a multi-omics study in a glioma mouse model revealed that glioma progression disrupts the gut microbiome, with a shift in bacterial composition; healthy mice had a dominance of Firmicutes, whereas glioma-bearing mice showed a prevalence of Bacteroides [56]. TMZ administration in the glioma-bearing mice led to significant changes in the gut microbiota composition and associated metabolites, suggesting a potential role of TMZ in restoring gut metabolic balance disrupted by glioma [56]. Additionally, research has shown that glioma development and TMZ treatment resulted in distinct alterations in fecal SCFAs and microbiome composition, indicating that TMZ may influence gut metabolite profiles affected by glioma [29]. These findings highlight the systemic effects of TMZ beyond its direct anti-tumor activity, particularly in mitigating glioma-induced changes in gut metabolites [57].

GBM creates an immunosuppressive environment, including high levels of TGF-β, which recruits Tregs and dampens effective anti-tumor immune responses [4]. This immune evasion further complicates treatment, making GBM one of the most challenging cancers to treat. Further, bacterial DNA has been detected in tumor tissues, suggesting that bacteria could have a more direct role in glioma biology by altering the nervous system microenvironment and tumor cell epigenetics [30,58]. The human tumor microbiome is composed of tumor-type-specific intracellular bacteria [59]. Mechanisms include inhibiting glioma cell migration and promoting tumor proliferation via the AHR pathway [35]. Imbalances in gut bacteria can disrupt microglial cell function and immune responses, influencing glioblastoma progression. Some studies even suggest that gut bacteria may influence the expression of immune checkpoint proteins like PD-L1 in GBM, potentially affecting the tumor’s ability to evade the immune system [47]. A study by Yan et al. employed Mendelian randomization analysis to examine the link between gut microbiota composition and the risk of developing GBM [60]. They identified 12 microbial groups significantly associated with GBM risk. Specifically, some bacteria such as Erysipelotrichaceae, Prevotellaceae, *Eubacterium nodatum*, Lachnospiridium, and Cyanobacteria were found to have a protective effect against GBM. In contrast, other groups like Rikenellaceae, Victivallaceae, *Ruminococcus gnavus*, Lactococcus, Ruminococcaceae, Sellimonas, and Desulfovibrionales were linked to an increased GBM risk. Among these, Lactococcus exhibited reverse causality, suggesting that the relationship between this genus and GBM could influence each other bidirectionally. Further analysis of blood samples through Mendelian randomization identified 19 metabolites potentially linked to a higher risk of GBM [60]. Notably, elevated levels of Pimeloylcarnitine/3-methyladipoylcarnitine were associated with GBM development. However, a significant reverse association was also observed, where the presence of GBM appeared to elevate these metabolite levels. The study highlighted a complex interplay between gut microbiota, metabolic changes, and glioblastoma development. By using the identified bacterial species in additional Mendelian randomization analyses, the researchers assessed the impact of these microbiota on the 19 metabolites, finding that eight microbial groups had a significant influence on the metabolite levels [60]. These findings suggest a multifaceted relationship between the gut microbiome, metabolites, and glioblastoma, providing potential avenues for understanding GBM pathogenesis and exploring new therapeutic targets (Figure 2).

## 5. Gut Microbiome Impacts Glioma Therapeutic Resistance and Progression

Altering gut microbiota with antibiotics in a syngeneic glioma mouse model increased glioma growth, likely through immune modulation. Antibiotic treatment changed gut microbiota composition, reduced cytotoxic NK cell populations, and altered microglial protein expression, highlighting the gut–immune axis as a potential factor in brain tumor progression [49]. Resistance mechanisms like the upregulation of efflux transporters such as P-glycoprotein (p-gp) at the BBB could be influenced by gut microbiota composition [3]. This could alter drug permeability and impact the effectiveness of chemotherapy in GBM patients. The BBB poses a significant obstacle in delivering therapeutic agents effectively to glioblastoma, limiting the efficacy of current treatments [5]. Temozolomide (TMZ) is favored over radiotherapy for treating glioblastoma due to its ability to cross the BBB and induce tumor cell death through DNA alkylation. Despite its widespread use, TMZ’s effectiveness is limited by resistance, which is often mediated by the DNA repair enzyme MGMT, resulting in less than 50% of patients responding to therapy [61].

The gut microbiome might influence this resistance by modulating the immune system. For example, Ruminococcaceae has been shown to alter the balance between M1 and M2 microglia potentially. A higher M2/M1 ratio is associated with worse outcomes in glioma, as M2 microglia are immunosuppressive and support tumor growth [48]. By influencing microglial polarization, Ruminococcaceae could shift this balance towards a more anti-tumor M1 profile, suggesting a potential therapeutic target. This has prompted research into strategies to increase TMZ sensitivity, including adjunct therapies like antibiotics that can alter the gut microbiome and influence the tumor microenvironment. Such approaches aim to optimize combination therapies and improve patient outcomes.

TMZ therapy improved the diversity and richness of gut microbiota in a glioma mouse model, reversing glioma-associated dysbiosis. Notably, the bacterial dominance shifted back from Bacteroides to Firmicutes, with significant changes observed in 17 bacterial genera and their associated metabolic pathways, including those involved in amino acid and lipid metabolism [56]. Certain bacteria, such as Bifidobacterium, might enhance TMZ’s efficacy through mechanisms like DNA methylation [56]. However, broad-spectrum antibiotics (ABX) can disrupt this beneficial relationship, as shown in studies where ABX administration reduced TMZ efficacy by depleting gut microbiota and diminishing immune responses like cytotoxic T cell recruitment [56]. This restoration of gut balance highlights TMZ’s role beyond its direct anti-tumor effects, suggesting a beneficial impact on gut microbiota.

Further studies revealed differences in the microbiome and metabolic profiles between TMZ-sensitive and non-sensitive individuals. While overall microbial diversity was similar, the specific composition, including the prevalence of Bacteroides, varied between groups. Distinct metabolic pathways, such as tryptophan metabolism, were altered, correlating with TMZ sensitivity. Additionally, immune profiling showed higher levels of immune markers IL-1β and TNF-α, along with greater infiltration of macrophages and cytotoxic T lymphocytes (CD8α), in TMZ-sensitive mice, suggesting that the reversal of immunosuppression is linked to better response to therapy. However, the use of broad-spectrum antibiotics (ABX) compromised the effectiveness of TMZ by disrupting the gut microbiota. ABX-treated mice displayed worsened glioma progression, decreased body weight, and increased tumor invasiveness, along with reduced immune cell infiltration in brain tissues. This suggests that a balanced microbiome is crucial for maintaining the immune responses that support TMZ’s anti-tumor activity. These findings underscore the role of gut microbiota in influencing TMZ efficacy and highlight the potential of microbiome-targeted strategies in optimizing glioblastoma treatment [56].

GBM is known to harbor a significant infiltration of glioma-associated macrophages and microglia (GAMs). Despite this, GBM is characterized by a “cold” tumor microenvironment due to its low mutational burden and minimal infiltration by cytotoxic T cells, which limits its immunogenicity [62]. Consequently, immunotherapy has not been a widely successful treatment strategy for glioblastoma, and the influence of gut microbiota on immunotherapy responses in GBM remains underexplored [63]. While studies in melanoma and pancreatic cancer have revealed marked differences in gut microbiota composition and diversity between responders and non-responders to anti-PD-1 therapy, such data are notably absent in glioblastoma due to its poor response to immunotherapy [64,65]. Preliminary research has investigated the potential of fecal microbiota transplantation (FMT) in GBM. In a study involving glioma-bearing mice, FMT from five healthy human donors was performed. Although no significant changes in glioma growth were observed between treated and control mice, the treated group demonstrated increased recruitment of cytotoxic T cells through IFN-γ activation, accompanied by a rise in the abundance of *Bacteroides cellulosilyticus* [66]. This finding underscores the need for further investigation into the immunomodulatory role of gut microbiota in GBM response to immunotherapy.

The critical role of gut microbiota in enhancing responses to immunotherapy in cancers such as melanoma and pancreatic cancer highlights the potential for exploring novel therapeutics that harness the gut microbiome to improve immunotherapy and radiotherapy outcomes in glioblastoma [67]. Radiotherapy, a cornerstone in GBM treatment, utilizes high-energy beams to target rapidly proliferating cancer cells, but it also risks collateral damage to normal host cells. Emerging evidence suggests that gut microbiota play a pivotal role in mediating both the therapeutic and adverse effects of radiotherapy. For example, short-chain fatty acids (SCFAs) and indole compounds exert immunomodulatory effects that enhance the anti-tumor response to radiotherapy [68]. In colorectal cancer, butyrate has been shown to improve radiosensitivity by modulating the transcriptional activity of Forkhead box class O3A (FOXO3A), a mechanism that could be further investigated in the context of GBM [69]. Although limited, existing research on the gut microbiota’s impact on radiotherapy effectiveness in glioblastoma warrants attention. Yang et al. conducted a study examining serum, peripheral blood mononuclear cells (PBMCs), and stool samples from glioma patients undergoing radiotherapy, stratified by rs4702 polymorphism [70]. Patients with the rs4702-A allele demonstrated increased Enterotype I and decreased Enterotype III in their gut microbiota, correlating with elevated expression of FURIN and brain-derived neurotrophic factor (BDNF) [70]. FURIN, a protease activating BDNF, is associated with enhanced cognitive functions and may mitigate radiation-induced cognitive decline [70]. These findings suggest a potential link between gut microbiota composition and radiotherapy outcomes, including cognitive preservation and cell cycle regulation. Furthermore, dietary interventions are increasingly recognized as a means of influencing the gut microbiome. Studies investigating restricted ketogenic diets in glioblastoma patients undergoing radiotherapy are underway, with a focus on evaluating not only the therapeutic outcomes but also their effects on the gut microbiota landscape [71]. These insights may offer novel avenues for optimizing radiotherapy responses through dietary modulation of the gut microbiome.

## 6. Insights into Modulating Gut Flora to Enhance Therapeutic Response in GBM

Recent research is exploring novel approaches that may enhance treatment outcomes. Immunotherapy, for instance, has shown promise in other cancers, but its success in GBM has been limited so far due to the brain’s unique immune environment and the tumor’s immunosuppressive microenvironment [72]. Immune checkpoint inhibitors, chimeric antigen receptor (CAR) T cell therapy, and oncolytic viruses are being investigated, though results have been modest [73,74].

Adjunct therapies aiming to modulate the gut microbiome hold promise for improving the response to TMZ and other treatments. Maintaining a balanced microbiome is critical for sustaining the immune environment required for effective treatment. The microbiome’s role is an emerging area of interest, as it has been shown to influence systemic immune responses. There is evidence that a healthy gut microbiome may enhance the efficacy of immunotherapies by modulating immune activity, and imbalances in the microbiome, or dysbiosis, may dampen immune responses. Researchers are investigating whether interventions that restore microbiome health could improve outcomes in GBM, especially by enhancing the effectiveness of immunotherapies. Interestingly, recent studies have shown that fecal microbial transplantation can optimize immunotherapy in GBM patients. More specifically, SCFAs can enhance the anti-tumor activity of CD8+ T cells by upregulating the expression of mTOR and the production of effector molecules such as CD25, IFN-γ, and TNF-α [75,76].

Ongoing research aims to better understand how the gut microbiome can be leveraged to optimize glioma therapy and predict patient outcomes [47]. Studies suggest that probiotics or dietary changes, like ketogenic diets, can alter gut microbiota and improve SCFA production, potentially influencing the CNS microenvironment and immune responses against GBM [4]. Fecal microbiota transplantation, an emerging therapy, is gaining popularity by suggesting that a balanced gut microbiome can restore homeostasis. Some research has explored fecal microbiota transplantation (FMT) as a method to restore gut balance in patients undergoing chemotherapy [75]. Zeng et al. utilized Mendelian randomization to explore the relationship between gut microbiota and GBM, identifying four taxa like Anaerostipes, Faecalibacterium, Prevotella7, and Ruminococcaceae as potentially protective against GBM [9]. Similarly, Anaerostipes has been recognized for its role in modulating the immune system to prevent colorectal cancer, highlighting its broader potential in anti-cancer therapy [77]. These findings suggest that targeted modulation of these bacterial populations could be a viable strategy for therapeutic intervention, potentially improving immune responses and reducing tumor burden.

Adjusting the gut microbiome with specific bacterial strains, like Faecalibacterium and Bifidobacterium, may enhance the effects of immunotherapies, such as PD-1 inhibitors, by modulating immune cell infiltration in tumors [78]. Studies have shown that microbiota diversity in glioma patients is significantly reduced compared to healthy individuals, suggesting a role for microbiota in the dysregulation of metabolic pathways and, ultimately, tumorigenesis [79]. Furthermore, the implementation of a ketogenic diet in glioma patients has demonstrated survival benefits, hinting at a possible role of the gut–brain axis in shaping the tumor microenvironment, with gut metabolites—and potentially gut flora—playing a key role [80]. D’Alessandro et al. also demonstrated that antibiotic treatment in glioma-bearing mouse models accelerated tumor growth, accompanied by alterations in natural killer cell and microglia phenotypes [49]. These findings open new avenues for utilizing gut flora as potential tools to enhance therapeutic responses in glioblastoma.

Promising research suggests that gut microbiota modulation via probiotics and fecal microbiota transplantation (FMT) may hold therapeutic potential. However, significant challenges remain, including the limited research on probiotics in glioblastoma and its lack of proven safety and efficacy. Recently, Windemuth et al. conducted preliminary studies optimizing probiotic delivery in orthotopic glioblastoma immunocompetent mouse models to evaluate their potential role in immunotherapy [81]. To mitigate inflammation induced by probiotic injection, an anti-VEGF antibody was administered prior to bacterial injection, successfully reducing bacteria-induced edema. This approach opens possibilities for stable colonization of glioblastoma mouse models to study probiotic mechanisms and therapeutic potential [81]. Additionally, an experimental study by Fatahi et al. explored the interaction between kefir and the U87 glioblastoma cell line, revealing a dose-dependent toxicity of kefir on GBM cells [82]. Furthermore, treatment of glioblastoma mouse models with a combination of probiotics *B. lactis* and *L. plantarum* was shown to suppress glioma growth by altering fecal metabolites and downregulating p-PI3K protein and survivin mRNA expression while upregulating *PTEN* mRNA and protein expression [83]. These cell cycle regulators are key components of the PI3K/AKT signaling pathway, which governs survival, proliferation, and metabolism—an essential pathway in glioblastoma tumor progression [84]. While some preliminary studies suggest that FMT may enhance immune responses in glioblastoma, research in this area remains in its infancy. For example, Fan et al. demonstrated that in glioma mouse models pre-treated with antibiotics, FMT slowed tumor growth [85]. However, the potential of FMT in immunotherapy has been more extensively explored in other cancers, such as melanoma. Phase I clinical trials have investigated the effects of healthy donor FMT combined with anti-PD-1 immunotherapy (nivolumab or pembrolizumab) in previously untreated melanoma patients. Responders to this treatment exhibited increased antigen-experienced cytotoxic T cell infiltration and decreased interleukin-8-expressing myeloid cells, which have been implicated in immunosuppression [86,87,88].

To further illustrate how gut microbiota influence therapeutic resistance in GBM and offers a promising target for treatment strategies, we have included the following Table 1:

## 7. Challenges and Future Directions

Despite promising insights, several challenges must be addressed before gut microbiota-based interventions can be integrated into glioblastoma treatment. The complexity of glioblastoma’s immunosuppressive microenvironment, inter-individual variations in gut microbiota, and the lack of standardized therapeutic protocols pose significant hurdles. Additionally, while gut microbiota have been shown to influence responses to immunotherapy and radiotherapy in other cancers, their role in glioblastoma remains poorly understood. Future research should focus on mechanistic studies, larger clinical trials, and personalized approaches to harness gut microbiota for therapeutic benefit in glioblastoma.

GBM treatment remains challenging due to the tumor’s complex biology and protective barriers like the BBB. However, innovative approaches involving the immune system, microbiome modulation, and improved drug delivery are being actively researched and hold promise for enhancing GBM therapy. Moving from animal models to human trials is crucial to validate findings about the gut–brain axis and GBM. This includes studies on microbiota-based biomarkers that could predict treatment responses [78]. Manipulating the microbiome in patients with weakened immune systems poses risks, such as infections or unanticipated immune reactions, which must be carefully considered in clinical applications [3]. Hence, the translational potential of gut microbiome-based therapies in GBM faces several critical challenges. The considerable variability in individual microbiomes and the lack of standardized methods for microbiome analysis complicate the development of universally applicable interventions. Additionally, the BBB presents a significant hurdle, limiting the effective delivery of microbial metabolites and immune modulators to the tumor. The complex interactions between the microbiome and GBM’s immune microenvironment remain poorly understood, further hindering the development of targeted therapeutic strategies. Furthermore, safety concerns and regulatory uncertainties continue to pose barriers to the clinical translation of microbiome-based treatments. Overcoming these challenges is essential to translating microbiome modulation from preclinical research into effective clinical applications for GBM therapy.

Nevertheless, the relationship between the gut microbiome and GBM presents a promising frontier in oncology. While current therapies remain limited in their efficacy, modulating the gut microbiome offers a new dimension for improving therapeutic responses and understanding GBM pathogenesis. Future research should focus on delineating the mechanisms by which gut bacteria influence GBM progression and exploring the potential of combining microbiome-targeted therapies with conventional treatments. Such advances could lead to personalized treatment strategies that harness the power of the microbiome to enhance patient outcomes in GBM. Emerging research highlights the gut microbiota’s role in GBM treatment, presenting both challenges and opportunities in personalized therapy. Compound K, a gut microbiota-derived metabolite, has demonstrated anti-migration properties in glioblastoma via the stromal cell-derived growth factor 1 (SDF-1), by reducing PKC and ERK phosphorylation, potentially serving as a predictive, prognostic, and diagnostic biomarker for therapy response [89,90]. Its ability to modulate tumor progression underscores the need for further investigation into microbiota-based therapeutic strategies. Surgical interventions further complicate the gut–glioma axis by significantly altering microbial diversity postoperatively, with microbial shifts linked to risk factors such as vessel injury and anesthesia-induced cardiovascular changes. Additionally, gut microbes influence glioma detection through NAD+ metabolism, aiding intraoperative imaging and surgical decision making [89]. Understanding these interactions may enhance preoperative assessment and postoperative recovery, reinforcing the importance of gut microbiota-targeted interventions in GBM management. Oncolytic viral therapies have also gained traction in glioblastoma treatment, with the gut microbiome emerging as a key player in enhancing viroimmunotherapy efficacy [91,92]. In a recent study, GSC-005 glioblastoma-bearing mice treated intratumorally with Delta-24-RGDOX, an engineered oncolytic adenovirus with an OX-40L expression cassette, exhibited distinct gut microbiota changes. Increased Bifidobacterium abundance correlated with improved survival, while CD4+ T cell depletion led to gut dysbiosis and reduced anti-tumor effects in untreated mice [93]. These findings suggest a strong interplay between the gut microbiome and viroimmunotherapy response, warranting further translational research.

The multifaceted potential of gut microbiota in GBM treatment is increasingly evident, with growing evidence supporting microbiota modulation as a means to improve treatment efficacy and survival outcomes. As research advances, targeting the gut–glioma axis may pave the way for novel therapeutic approaches, optimizing both surgical and immunotherapeutic interventions in GBM patients.

## Figures and Tables

**Figure 1 ijms-26-02935-f001:**
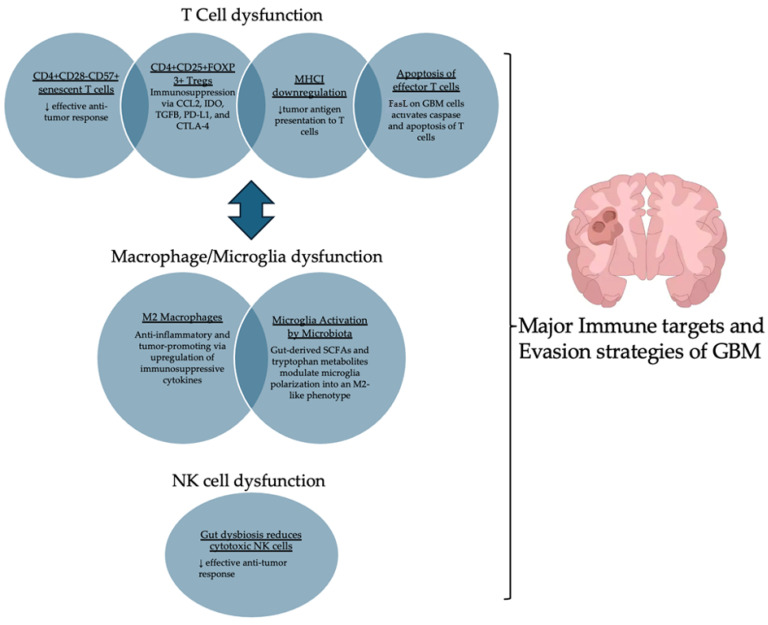
Schematic representation of immune dysfunction in GBM. This figure illustrates how GBM suppresses the immune system by impairing T cell function, NK cell function, and inducing macrophage polarization. The arrows depict the interdependence between immunosuppressive pathways, particularly the feedback loop between Tregs and M2 macrophages, emphasizing how GBM reprograms the immune microenvironment to evade anti-tumor immunity.

**Figure 2 ijms-26-02935-f002:**
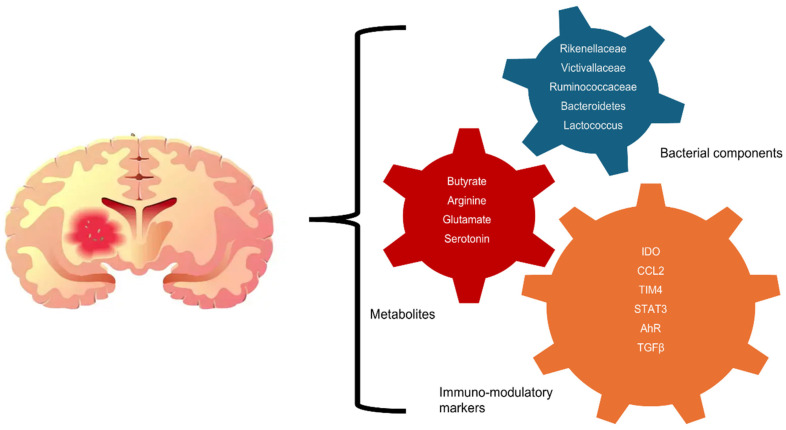
Schematic representation of gut microbiota, metabolites, and immune modulators in glioblastoma progression, illustrating the role of different bacterial groups, metabolites, and immune modulators in promoting glioblastoma (GBM) progression. The interconnected gears symbolize the dynamic and complex interplay between these factors, even though direct evidence for their interactions remains to be proven. Blue gear (gut bacterial components): highlights bacterial groups such as *Rikenellaceae*, *Victivallaceae*, *Ruminococcaceae*, and *Lactococcus*, which influence immune responses and tumor progression through their interactions with the immune system and metabolites. Red gear (metabolites): showcases key metabolites, including butyrate, arginine, glutamate, serotonin, and dopamine. These metabolites are critical for immune regulation, tumor metabolism, and modulation of the tumor microenvironment. Orange gear (immune modulators): highlights immunosuppressive factors such as IDO, CCL2, TIM4, STAT3, TGF-β, AhR, CTLA-4, and PD-L1, which enable glioblastoma cells to evade immune detection and foster an immunosuppressive environment.

**Table 1 ijms-26-02935-t001:** Comparative summary of gut microbiota’s impact on GBM therapy response. This table summarizes key interactions between the gut microbiota and GBM, focusing on immunomodulation and therapeutic implications. It highlights specific microbiota-derived metabolites, immune system interactions, and potential therapeutic strategies.

Factor	Mechanism	Effect on GBM	Implication for Therapy	References
Microbiota Diversity	↓ Diversity in GBM patients, shift toward pro-inflammatory taxa	Promotes immunosuppressive tumor microenvironment (TME), favors MDSCs and Tregs	Microbiome-targeted interventions may enhance response to immunotherapy	[7,55,75]
SCFA Production (Butyrate, Propionate, Acetate)	Modulates Treg and Th1/Th17 balance, increases antigen presentation by DCs	Enhances anti-tumor immunity, reduces chronic inflammation	Could enhance TMZ/radiotherapy efficacy by improving immune activation	[4,67,68]
Tryptophan Metabolism (Kynurenine Pathway Shift)	↑ IDO1-mediated kynurenine production ↑ Treg, ↓ cytotoxic CD8+ T cells	Suppresses anti-tumor immune responses, promotes T cell exhaustion	IDO1 inhibition could improve response to checkpoint inhibitors	[39,55,62]
Microbiota-Mediated BBB Modulation	Alters P-glycoprotein (P-gp) and tight junction expression, affecting drug penetration	Impacts CNS drug bioavailability and immune cell infiltration	Targeting microbiota to regulate BBB permeability could improve chemotherapy efficacy	[3,72,77]
FMT Studies (Mouse Models)	Restores gut microbiome balance, improves gut--immune crosstalk	Enhances response to immune checkpoint blockade (ICB)	Potential adjunct for GBM immunotherapy	[65,75]
Probiotic Intervention (Lactobacillus, Bifidobacterium)	Modifies dendritic cell function, increases IL-12 and IFN-γ	Boosts anti-tumor immunity but effects in GBM remain unclear	May support immune checkpoint therapy but requires trials	[80,81,89]

## Data Availability

Not applicable.

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
