# Peer review of "The Role of Gut Microbiome on Glioblastoma Oncogenesis and Malignant Evolution"

_ijms, 2025, doi:10.3390/ijms26072935_

Round 1

Reviewer 1 Report

Comments and Suggestions for Authors

The review article by Zaynab Sidi Mohamed et al., entitled "The Role of Gut Microbiome on Glioblastoma Oncogenesis and Malignant Evolution" discusses the oncogenic role of gut microbiota in GBM, particularly its impact on therapeutic resistance and tumour progression.
The article's objective is clearly defined. The 'Introduction' section is well prepared and very informative, providing detailed information about glioblastoma and the importance of microbiota. The subsequent chapters offer a comprehensive overview of the gut microbiome in cancer and provide detailed insights into the potential link between the gut microbiome and various cancers.The next three chapters focus on glioblastoma, particularly therapeutic resistance, progression, and therapeutic response. However, the chapters are written in such a detailed manner that they are somewhat challenging to comprehend. There is only one figure that summarizes the gut microbiota, metabolites, and immune modulators in glioblastoma progression. The references are generally relevant, up-to-date and correctly referenced. The manuscript addresses a very important issue and has huge potential, but it is a little chaotic. More figures and tables would provide the same information in a more consistent and ordered way.

Minor comments:

  1. The whole manuscript needs to be checked for font size and type. The same goes for citations through the text. Please be consistent in style.
  2. References – number 75 needs complementing.
  3. The whole manuscript must be better organised to enable better understanding of the topic.
  4. There is a need to add more figures and tables to make it more attractive for the reader and better understandable.

Author Response

Reviewer 1

Opening comment: The review article by Zaynab Sidi Mohamed et al., entitled "The Role of Gut Microbiome on Glioblastoma Oncogenesis and Malignant Evolution" discusses the oncogenic role of gut microbiota in GBM, particularly its impact on therapeutic resistance and tumour progression. The article's objective is clearly defined. The 'Introduction' section is well prepared and very informative, providing detailed information about glioblastoma and the importance of microbiota. The subsequent chapters offer a comprehensive overview of the gut microbiome in cancer and provide detailed insights into the potential link between the gut microbiome and various cancers.The next three chapters focus on glioblastoma, particularly therapeutic resistance, progression, and therapeutic response. However, the chapters are written in such a detailed manner that they are somewhat challenging to comprehend. There is only one figure that summarizes the gut microbiota, metabolites, and immune modulators in glioblastoma progression. The references are generally relevant, up-to-date and correctly referenced. The manuscript addresses a very important issue and has huge potential, but it is a little chaotic. More figures and tables would provide the same information in a more consistent and ordered way.”

Response: Thank you kindly for taking the time to read and comment our manuscript.

Minor comments:

  1. The whole manuscript needs to be checked for font size and type. The same goes for citations through the text. Please be consistent in style.

Response: We appreciate the reviewer for this input. We have standardized formatting and references: font size, citation styles, and reference completeness have been carefully reviewed and standardized throughout the manuscript.

  1. References – number 75 needs complementing.

Response: We appreciate the suggestion and Reference 75 has been updated with new citations which would be more appropriate.

  1. The whole manuscript must be better organised to enable better understanding of the topic.

Response: We appreciate the suggestion and have improved manuscript structure: streamlined transitions between sections to ensure a more natural flow of ideas. Several summary sentences were added where necessary to improve readability. We also added a roadmap sentence in the Introduction: A clear roadmap paragraph now explicitly outlines the structure of the review, preemptively addressing concerns about organization and logical flow.

  1. There is a need to add more figures and tables to make it more attractive for the reader and better understandable.

Response: We thank the reviewer for this great suggestion. We have incorporated an additional figure (New Fig.1) illustrating immune failure in GBM and one summary table (Table 1) detailing the gut microbiome’s role in therapeutic resistance and immune modulation.

Reviewer 2 Report

Comments and Suggestions for Authors

Mohamed et al. present a unique review on the role of the gut microbiome on GBM. It covers relevant bacterial phyla, including Firmicutes, Bacteroidetes, Actinobacteria, Proteobacteria, Fusobacteria, and Verrucomicrobia. It very clearly leads a reviewer through the microbiome in cancers and then focuses on Glioma Development. It then further elevates the field by delving into the mechanisms whereby the Gut Microbome alters the efficacy of therapeutics. Figure 1 is quite innovative. It finalizes the review with a very nice conclusion and future directions section, particularly in covering clinical translation. The review is well written and comprehensive. The references are appropriate. Overall, this timely work highlights important and novel aspects of this relationship and there seems to be a deficiency in current review literature regarding this topic. The author's should be commended on their work.
Minor:
1)      I'm not sure, but it looks like the abstract has different fonts? It may just be the PDF but double-check. Also, some of the reference numbers are clearly not the same size as the letters (ex; 46, 9, 51).
2)      Citation needed for line. "In addition to dietary habits, the mode of delivery (vaginal birth vs. C-section) has a long-term impact on an individual's microbiota."
3)      There's a couple of places throughout the document where the font does look different, both the size and font itself.

Author Response

Reviewer 2

Opening comment: Mohamed et al. present a unique review on the role of the gut microbiome on GBM. It covers relevant bacterial phyla, including Firmicutes, Bacteroidetes, Actinobacteria, Proteobacteria, Fusobacteria, and Verrucomicrobia. It very clearly leads a reviewer through the microbiome in cancers and then focuses on Glioma Development. It then further elevates the field by delving into the mechanisms whereby the Gut Microbome alters the efficacy of therapeutics. Figure 1 is quite innovative. It finalizes the review with a very nice conclusion and future directions section, particularly in covering clinical translation. The review is well written and comprehensive. The references are appropriate. Overall, this timely work highlights important and novel aspects of this relationship and there seems to be a deficiency in current review literature regarding this topic. The author's should be commended on their work.

Response: We appreciate the reviewer for taking the time to provide these constructive comments which we tried to address as best as we could to improve our manuscript.

Minor:
1. I'm not sure, but it looks like the abstract has different fonts? It may just be the PDF but double-check. Also, some of the reference numbers are clearly not the same size as the letters (ex; 46, 9, 51).

Response: We appreciate the suggestion and adjusted the font in the abstract and references.

2. Citation needed for line. "In addition to dietary habits, the mode of delivery (vaginal birth vs. C-section) has a long-term impact on an individual's microbiota."

Response: Thank you for the comment. We added a reference (ref.16) for the C-section vs vaginal delivery impact on gut microbiota population.

3. There's a couple of places throughout the document where the font does look different, both the size and font itself.

Response: We appreciate the comment. We have adjusted all the size and font and made them consistent.

Reviewer 3 Report

Comments and Suggestions for Authors

This review provides a comprehensive analysis of research achievements on the intersectionality between the gut microbiome and GBM. The topic is interesting. The structure and statement of this review. The paper needs some minor revisions before acceptance.

  1. It is suggested to check the numbering and labeling of references, such as, reference (3) (4).
  2. It is suggested that the reference titles use the same format, for example, reference 15, 16.

Author Response

Reviewer 3

Opening comment: This review provides a comprehensive analysis of research achievements on the intersectionality between the gut microbiome and GBM. The topic is interesting. The structure and statement of this review. The paper needs some minor revisions before acceptance.

Response: Thank you kindly for taking the time to read and comment our manuscript.

1. It is suggested to check the numbering and labeling of references, such as, reference (3) (4).

Response: We appreciate the reviewer for this input. We have checked all the numbering and labeling of references including font size, citation styles, etc.

2. It is suggested that the reference titles use the same format, for example, reference 15, 16.

Response: We appreciate the suggestion and adjusted the format of all the references.

Round 2

Reviewer 1 Report

Comments and Suggestions for Authors

Suggested corrections have been made. The only comment currently is Figure 1 - it does not fit on the page. It also has visible red highlights left on it.